# Longitudinal Outcomes of Gender Identity in Children (LOGIC): protocol for a prospective longitudinal cohort study of children referred to the UK gender identity development service

Eilis Kennedy ,[1,2,3] Lauren Spinner,[1] Chloe Lane,[1] Hannah Stynes,[1] Veronica Ranieri ,[1,2] Polly Carmichael,[4] Rumana Omar,[5] Victoria Vickerstaff,[6,7] Rachael Hunter,[6] Talen Wright,[7] Robert Senior,[1,3] Gary Butler,[4,8] Simon Baron-Cohen,[9] Bridget Young,[10] Michael King[7]

For numbered affiliations see end of article.

**Correspondence to**
Dr Eilis Kennedy;
ekennedy@tavi-port.nhs.uk

## ABSTRACT

**Introduction** Gender identity development services (GIDS) worldwide have seen a significant increase in referrals in recent years. Many of these referrals consist of children and young people (CYP) who experience gender-related distress. This study aims to improve understanding of outcomes of CYP referred to the UK GIDS, specifically regarding gender identity, mental health, physical health and quality of life. The impact of factors such as co-occurring autism and early social transition on outcomes over time will be explored.

**Methods and analysis** This is a prospective cohort study of CYP aged 3–14 years when referred to the UK GIDS. Eligible participants will be ≤14 years at the time their referral was accepted and will be on the waitlist for the service when baseline measures are completed. Children aged under 12 years will complete the measures in an interview format with a researcher, while young people aged 12 years and over and their parents/caregivers will complete online or paper-based questionnaires. Participants will complete follow-up measures 12 months and 24 months later. The final sample size is expected to be approximately 500. Logistic regression models will be used to explore associations between prespecified explanatory variables and gender dysphoria. Appropriate regression models will also be used to investigate explanatory variables for other outcomes. Subgroup analyses based on birth-assigned gender, age at referral and co-occurring autistic traits will be explored.

**Ethics and dissemination** The study has been approved by the Health Research Authority and London – Hampstead Research Ethics Committee (reference: 19/LO/0857). The study findings will be published in peer-reviewed journals and presented at both conferences and stakeholder events. Findings will be used to inform clinical practice.

## INTRODUCTION

Gender identity development services (GIDS) for children and young people (CYP) have experienced a significant rise

### Strengths and limitations of this study

► This study will recruit a large cohort from one of the world's largest gender identity development services (GIDS) for children and young people.

► The prospective longitudinal cohort study design will enable documentation of the natural history and outcomes over time of children and young people referred to the UK GIDS.

► Data will be collected directly from both children and young people and their parent/caregiver.

► As this is a 2-year prospective study, it will not be possible to ascertain longer term health and well-being outcomes and treatment decisions.

► As the study is recruiting families on the GIDS waitlist, this is a potentially challenging population to access as they are not in receipt of clinical services at the time of recruitment.

in referrals over recent years.[1 2] In the UK GIDS, this is evidenced by an increase from 1408 referrals of CYP in 2015/2016 to 2728 referrals of CYP in 2019/2020.[3] A changing pattern in referrals has also been observed, and in particular, there has been a shift to an increase in referrals of CYP assigned female at birth.[4 5] It has been noted that CYP referred to the GIDS are a heterogeneous group, and while some report distress associated with a mismatch between their gender assigned at birth and their gender identity, others do not report such distress.[6] Furthermore, some CYP who are referred to the service identify with a non-binary identity.[6 7] Although these studies provide important insights into the characteristics of CYP who are referred to the GIDS, large-scale studies assessing outcomes over time for these CYP are lacking, and thus,

longitudinal research is needed to further inform clinical practice.

Findings from a number of studies indicate that CYP referred to gender services often experience mental health difficulties, such as depression and anxiety, with prevalence tending to increase with age.[8–10] High rates of self-harm and suicidality are also common.[10–12] Furthermore, co-occurring gender dysphoria and autism have been reported for some CYP who are referred to gender services.[13 14] As co-occurring conditions are frequently reported for gender diverse CYP, it is important to explore the extent to which these impact longer term outcomes and to consider how clinical services can best support the needs of these CYP.

An increasing number of younger children who have already socially transitioned (ie, changed clothing, hairstyle, first name and pronouns to reflect their gender identity) at the time of referral to gender services has been reported.[15] Recent research has suggested that this early social transition can have desirable outcomes for CYP, particularly in relation to a reduction in mental health difficulties.[16 17] However, the longer term impacts of early social transition in childhood have not yet been determined.[18 19] Thus, prospective longitudinal research is needed to investigate outcomes in relation to early social transition, particularly in prepubertal children.[20]

Current NHS intervention for CYP experiencing gender dysphoria is aimed at alleviating dysphoric feelings, improving well-being and supporting CYP and their families to make informed decisions about treatment and the meaning of gender identity within their own lives and contexts.[6] However, the evidence on which current treatment protocols are based is widely acknowledged to be limited, and predictors of individual gender and psychosexual developmental pathways are not clear.[18] It can therefore be difficult to know which CYP referred to GIDS are more likely to access medical treatment (hormone blockers to suppress puberty or cross sex hormones). In addition, factors that may be associated with the likelihood of receiving such treatment and the physical and psychological impacts of receiving this treatment have not yet been fully ascertained.[21] Identifying these factors should provide much needed evidence to inform the care of gender diverse CYP attending services.

There is also limited research on the direct healthcare costs and indirect wider societal costs for CYP referred to gender services and this information is important for policy makers and service providers to consider when planning services. In the context of a finite health budget, reporting this information alongside health and quality of life gains provides information on the potential value for money of gender services and how they might best be configured for maximum benefit to the healthcare service and wider society.

In summary, this study aims to improve understanding of the outcomes of CYP referred to the UK GIDS, specifically regarding gender identity, mental health, physical health and quality of life. The impact of factors such as co-occurring autism and early social transition on outcomes over time will be explored. The UK GIDS is one of the largest specialist gender services for CYP in the world and is therefore an ideal service from which to recruit. This is the first longitudinal study to assess outcomes of CYP referred to gender services in the UK and is novel in terms of recruiting families who are on the waitlist for the GIDS at baseline, the inclusion of a cohort of CYP who were ≤14 years at the time of referral and the commitment to follow-up over time, regardless of whether families actually attend the service.

## Aims
This study aims to investigate the outcomes of CYP referred to the UK GIDS and the factors that are associated with these outcomes. Specifically, the study aims to establish: (1) the proportion of CYP who experience ongoing gender dysphoria and access medical treatment; (2) factors that are associated with ongoing gender dysphoria and treatment choice; and (3) how (A) medical treatment, (B) social transition and (C) co-occurring autism impact physical health, well-being, quality of life, peer and family relationships and cost to the National Health Service (NHS) and other public services.

## METHODS AND ANALYSIS
### Study design
A prospective longitudinal cohort study examining the outcomes of CYP referred to the UK GIDS. This study forms part of a wider programme of research investigating Longitudinal Outcomes of Gender Identity in Children (the LOGIC study).[22] This programme of research uses a mixed methods approach, incorporating both quantitative and qualitative longitudinal studies to investigate the experiences, outcomes and well-being of families referred to the UK GIDS. Please see the LOGIC-Q protocol for details on the longitudinal qualitative study, which involves semistructured interviews with a purposively sampled subset of up to 40 families who have completed the quantitative assessments (McKay *et al*, under review).

### Study setting
The study is based at the Tavistock & Portman NHS Foundation Trust and is recruiting participants from the GIDS waitlist. The GIDS is a nationally commissioned service covering England, Wales, Northern Ireland and in part, Scotland and the Republic of Ireland, through a series of outreach clinics and main hubs in London and Leeds. The study start date was July 2019, and the anticipated end date is May 2023.

### Participants and eligibility
Participants will consist of CYP aged 3–14 years at the time their referral to the GIDS was accepted and their parent/caregiver. Referrals can be made by Child and Adolescent Mental Health Services, or from other health, social care or educational professionals, including GPs. The service

does not accept self-referrals from CYP or their families. In order for the referral to be accepted, the referral must indicate that the CYP is experiencing concerns or distress around their gender identity. Inclusion criteria for CYP are: had a referral to the GIDS accepted; awaiting first appointment with the service when baseline measures are completed; speak English; live in the UK or the Republic of Ireland. Families will still be eligible to participate in the study even if, after referral, they no longer wish to remain on the waitlist for an appointment with the GIDS. Exclusion criteria for CYP are: aged >14 years at the time their referral to the GIDS was accepted; already attended an appointment at the GIDS before baseline measures are completed; do not provide assent to take part; and do not have parent/caregiver consent. Parents/caregivers of eligible CYP will be asked to provide informed consent in order to participate and must also be proficient in English and live within the UK or the Republic of Ireland.

### Recruitment and consent

Potential participants will be identified from referrals received by the GIDS, and recruitment will consist of a three-step process. Potential participants (CYP and their families) will receive a study invitation letter informing them of the study and inviting them to take part. At least three working days after the letter is sent, the family will be contacted by telephone by a member of the GIDS administrative team. The purpose of this telephone call is to highlight the study to potential participants and seek their consent to pass their contact details to the LOGIC research team. If potential participants agree for their contact details to be passed on, a researcher from the LOGIC research team will then contact them directly and provide them with information sheets, including an age-appropriate version for the CYP. Families will have the opportunity to discuss the study with the researcher, and if they are happy to participate, arrangements will be made for completion of the baseline measures.

Informed consent will be requested from all parents/caregivers at baseline prior to completion of the assessment measures. All CYP will be asked to assent at baseline and follow-up time points. Consent will be requested and recorded on either a paper or online consent form, depending on how the measures are completed. Participants will be informed that their decision to participate in the study (or not) will have no impact on their access to care, their position on the referral waiting list or their future care pathway. Any clinical care team that the participant may go on to have will not know that the individual is taking part in the study unless they choose to share that information with them. Participation in the study will not be routinely documented on the participant's medical record. The only exception to this is if risk is identified for the child or young person, in which case the research team will follow a risk protocol that may result in a referral to the GIDS enquiries line to initiate support for the family. This information would be documented on the individual's medical record, but referrals

such as this will only be made with the parent/caregiver's consent. Participants will be informed that they have the right to withdraw from the study at any time, without providing a reason and without any impact on their access to care or their care pathway.

### Data collection and measures

Data will be collected from both the CYP and their parent/caregiver. However, in some circumstances (eg, CYP does not wish to participate), data may only be collected from a parent/caregiver. Depending on the age of the CYP and participant preference, assessments will be carried out either in a face-to-face, video call or telephone interview format with a research assistant (this method will primarily be used with children <12 years and/or those with neurodevelopmental conditions) or via a self-report questionnaire in a paper or electronic format (via Qualtrics, an online survey platform). It is anticipated that the majority of participants aged ≥12 years and their parent/caregiver will complete the questionnaires online via the Qualtrics survey platform. In the event that a participant becomes distressed or discloses a risk/safeguarding concern during an assessment, the researchers will adhere to the study risk protocol.

Questionnaires and assessments will include measures of: gender dysphoria and gender identity; social transition; mental health and well-being; autistic traits; self-evaluation of puberty stage, height and weight; quality of life; peer and family relationships; sociocultural and demographic factors; and healthcare and societal costs. Measures will be completed at baseline and 1-year and 2-year follow-up periods. For a detailed list of measures, please see table 1.

For participants who are referred to and attend an endocrinology clinic at University College London Hospitals NHS Foundation Trust or Leeds Teaching Hospitals NHS Trust, access to routinely collected clinical measurements of body composition, bone density and age and blood test values will be requested. Participants will be given the option to consent to the research team accessing this information, and therefore, requests will only be made for participants who provide consent. Clinicians and/or endocrinologists will not know who is taking part in the study or for which participants this information is requested. These data will be collected by a member of the LOGIC research team with assistance from a research nurse at each site. As per current NHS treatment protocols, those receiving medical treatment with hormone blockers (gonadotropin-releasing hormone agonists (GnRHa)) and/or cross-sex hormones will be monitored for changes in blood count, renal and liver function, skeletal maturity and bone density and Tanner pubertal staging. These data will allow adult height predictions to be tracked and clinical measurements of body proportions, height, spinal height, shoulder and hip widths and body composition by bioelectrical impedance to be taken. For a list of these clinic derived variables, please see table 2.

**Table 1** Summary of assessment measures to be completed at each time point

| Measure | Data collected | Informant/source | Baseline | 12 months | 24 months |
|---|---|---|---|---|---|
| **Demographics/ background** | Caregiver: age, gender, education, occupation, sexual orientation, ethnicity and relationship to child.<br>CYP: age, sex assigned at birth, ethnicity, age at referral and background to referral. | P | × | × | × |
| **Gender identity** | | | | | |
| Parent Questionnaire | DSM-5 criteria for gender dysphoria in child; family experiences of child's gender identity; gender expression and social transition; and body image and body distress. | P | × | × | × |
| Child Semi-Structured Interview | Verbal identification with experienced gender; gender expression and social transition; age first realised and expressed gender identity; and body image and distress. | C | × | × | × |
| Young Person Questionnaire | DSM-5 criteria for gender dysphoria; distress around gender identity; gender expression and social transition; body image and body distress; and sexual orientation. | YP | × | × | × |
| Gender Similarity Task[28] | Perceived similarity to own-gender and other-gender peers. | C | × | × | × |
| Gender Diversity Questionnaire[7] | Verbal identification with expressed gender; gender fluidity; gender expression; age first realised and expressed gender identity. | YP | × | × | × |
| Utrecht Gender Dysphoria Scale[29] | Severity of gender dysphoria. | YP | × | × | × |
| Gender Identity Self Stigma Scale[30] | Internalised transphobia and stigma. | YP | × | × | × |
| **CYP functioning** | | | | | |
| Child Behaviour Checklist[31] | Emotional reactivity; anxious/depressed; somatic complaints; withdrawn; sleep problems; attention problems; and aggressive behaviour. | P | × | × | × |
| Autism Quotient[32] | Autistic traits in child or adolescent. | P | × | × | × |
| Strengths and Difficulties Questionnaire[33] | Emotional symptoms; conduct problems; hyperactivity/inattention; peer relationship problems; and prosocial behaviour. | P and YP | × | × | × |
| Youth Self Report[31] | Anxious-depressed; withdrawn/depressed; somatic complaints; social problems; thought problems; attention problems; rule-breaking behaviour; and aggressive behaviour. | YP | × | × | × |
| Moods and Feelings Questionnaire[34] | Depressive symptoms in child or adolescent. | YP | × | × | × |
| Warwick-Edinburgh Mental Well-being Scale[35] | Mental well-being. | YP | × | × | × |
| Child and Youth Resilience Measure[36] | Brief screener for resilience processes in the lives of children and adolescents. | C and YP | × | × | × |
| **Healthcare resource use** | | | | | |
| Child & Adolescent Service Use Schedule[37] | All-cause uses of hospital, community-based and private health and social care services. | P | × | × | × |
| **Quality of life** | | | | | |

Continued

**Table 1** Continued

| Measure | Data collected | Informant/source | Baseline | 12 months | 24 months |
|---|---|---|---|---|---|
| Kidscreen-52[38] | Physical well-being, psychological well-being, moods and emotions, self-perception, autonomy, parent relations and home life, social support and peers, school environment, social acceptance (bullying), and financial resources. | P or C, YP | × | × | × |
| Child Health Utility 9 Dimensions[39] | Paediatric health-related quality of life for use in economic evaluation (quality adjusted life years). | P or C, YP | × | × | × |
| **Physical measures** | | | | | |
| Parental height | Height in centimetres. | P | × | | |
| CYP pubertal development | Tanner stage. | P and YP | × | × | × |
| CYP height, weight, BMI | Height in centimetres; weight in kilograms. | P and YP | × | × | × |

Child (C) ages for data collection are 3–11 years. Young people (YP) ages for data collection are ≥12 years. Parental (P) proxy measures are available for children (aged <8 years) not able to self-complete quality of life instruments.
BMI, body mass index; CYP, children and young people.

**Table 2** Endocrinology clinic derived measurements for young people receiving medical treatment

| | Data source | Initial assessment | Follow-up (FU) 3–6 months on GnRHa | Follow-up (FU) 12 months on GnRHa (monitoring cycle repeats) |
|---|---|---|---|---|
| **Auxology/physical measures** | | | | |
| Height, sitting height, biacromial width, bi-iliac width and weight | Clinical measures | × | × | × |
| BMI | Value derived by calculation | × | × | × |
| Parent's height | Clinical measure | × | | |
| Pubertal Tanner staging | Clinical examination | × | × | × |
| Body composition assessment | Tanita bioelectrical impedance analyser | × | × | × |
| **Radiological measures** | | | | |
| Bone densitometry | Dexa scan | × | × | × |
| Bone age | X-ray of left hand and wrist | × | × | × |
| **General blood measures** | | | | |
| Full blood count, ferritin (iron) | Blood test | × | × | × |
| Renal, liver, lipid and bone profiles | Blood test | × | × | × |
| Vitamin D | Blood test | × | × | × |
| **Endocrine measures** | | | | |
| Thyroid function test | Blood test | × | × | × |
| Reproductive hormones: FSH, LH, PRL, testosterone, oestradiol | Blood test | × | × | × |

BMI, body mass index; FSH, follicle stimulating hormone; GnRHa, gonadotropin-releasing hormone agonists; LH, luteinising hormone; PRL, prolactin.

Personal data provided by potential participants during the recruitment phase, prior to assessments, will be held in a password-protected database on an encrypted NHS server at the Tavistock and Portman NHS Foundation Trust, accessible solely to the chief investigator and immediate research team. Pseudonyms, in the form of participant ID numbers, will be assigned to participants once they have agreed to complete the baseline measures. Participant ID numbers will be held in a password-protected database on an encrypted NHS server at the Tavistock and Portman NHS Foundation Trust.

## Sample size

It is estimated that 638 participants will be recruited over an 18-month period assuming a recruitment rate of 70% of all eligible referrals of CYP to the GIDS. Allowing for 20% attrition, the final sample size with complete follow-up data is expected to be 510 participants. Assuming the proportion of CYP with gender dysphoria to be 40%, a sample size of 510 (number of GD events=204) will allow us to estimate 20 regression coefficients with adequate precision using a logistic regression model, based on prespecified explanatory variables.[23] This sample size is more than sufficient to investigate the association between these explanatory variables and other continuous clinical outcomes using a linear regression model.[24] For the analysis of the binary gender dysphoria outcome, if we are unable to obtain complete follow-up data from 510 participants, we will limit the number of explanatory variables that will be included in the multivariable regression model. This decision will be based on the clinical importance/relevance of the variables, and this will be prespecified in the statistical analysis plan.

During baseline recruitment, as our initial target of recruiting 70% of incoming referrals of CYP aged ≤13 years was not being achieved, our recruitment strategy was reviewed, and the decision was made to also recruit families on the existing GIDS waitlist and to increase the eligibility criteria to those aged ≤14 years when their referral to the GIDS was accepted. The families of all CYP who meet the eligibility criteria will be invited to participate in the study.

## Analysis plan

Data will be analysed in STATA by the statistical team at UCL PRIMENT's Clinical Trials Unit. Pseudonymised datasets will be sent to the team via University College London's (UCL) General Data Protection Regulation (GDPR) compliant Data Safe Haven file transfer portal. Participant characteristics at baseline and outcomes will be presented using mean (SD), median (IQR) or frequencies (proportion), as appropriate. Characteristics and outcomes of participants who drop out of the study will be described and compared descriptively to those with complete follow-up data in order to establish whether there are differences between these participants or factors that may have motivated them to leave the study.

A logistic regression model will be used to explore associations between the prespecified explanatory variables and the binary gender dysphoria outcome. Linear regression models will be used for the continuous clinical outcomes. Appropriate regression models will be used to investigate explanatory variables for other outcomes. Models that account for repeated measurement data will be used where appropriate, including time period as an additional covariate. Model assumptions regarding linear relationships and normality for continuous variables will be investigated. Any violation of these assumptions will be handled using appropriate statistical methods. Exploratory subgroup analyses will be carried out for prepubertal children and CYP with autistic traits using descriptive statistics and appropriate regression models, where possible. Physical health, well-being and quality of life outcomes over time will be compared descriptively between those in receipt of hormone blockers (GnRHa) and those of a similar age/pubertal stage who progress through puberty without such intervention. Physical health data routinely collected by the endocrine service (see table 2) will be reported descriptively.

Resource use will be collected using the child and adolescent services use schedule (CA-SUS), which has been specifically adapted for this study. The CA-SUS collects information about service use in the past 6 months regarding heath care resource use, including: GIDS, private, primary and secondary care, mental health, school attendance, educational support, voluntary services and medication. Reponses to the CA-SUS will be costed using the Personal Social Services Resource Unit,[25] NHS reference costs,[26] British National Formulary[27] and other published sources. Descriptive statistics for resource use and cost (numbers, percentages, means and SD per CYP) will be reported by CYP in active contact with the service versus those who are not. Further subgroup analyses will explore resource use and cost by medical treatment, social transition, co-occurring autistic traits and prepubertal children. Key predictors of costs will be investigated by including prespecified explanatory variables and 'active contact with GIDS' in appropriate general linear models (GLMs). The most appropriate GLM will be chosen using the Akaike information criteria. This will allow additional cost or cost-savings associated with active GIDS contact to be estimated. Similar models will also be used to investigate whether active contact with GIDS is related to improvements in quality of life. This will be based on quality-adjusted life years calculated using the CHU9D, the relevant tariff and the area under the curve.

Bias in relation to missing data will be investigated. If required, logistic regression will be used to identify predictors of missingness, and these predictors will be included in the analysis models as part of secondary analyses. Multiple imputation may be used to impute the missing values of the explanatory variables, if appropriate.

## Data storage

All data will be held in a password-protected database on an encrypted NHS server at the Tavistock and Portman NHS Foundation Trust. Data processing agreements and data sharing agreements are in place between the Tavistock & Portman NHS Foundation Trust and study collaborators. Data transferred to UCL for analysis will be retained in a GDPR-compliant Data Safe Haven, which is security key and password-protected.

## Patient and public involvement

The LOGIC study was developed in collaboration with a service user coapplicant and coauthor (TW), UK GIDS users and external stakeholders. CYP and their families using the UK GIDS were consulted regarding their priorities for research at GID Service User Family Days in London and Leeds. This informed the research proposal and application for funding, which was shared and further reviewed with three parents of young people accessing the service. Throughout the study, the research team will host CYP and parent/caregiver advisory groups who will meet every 6 months to provide ongoing feedback and advice on the study. These will include families who are participating in the LOGIC study.

## ETHICS AND DISSEMINATION

This study has been approved by the Health Research Authority and London – Hampstead Research Ethics Committee (Reference: 19/LO/0857). The study findings will be published in peer-reviewed journals and presented at both conferences and stakeholder events. No identifiable data will feature in the dissemination of results. It is hoped that the findings will be used to inform clinical practice in the NHS and elsewhere.

**Author affiliations**
[1]Research & Development Unit, Tavistock and Portman NHS Foundation Trust, London, UK
[2]Research Department of Clinical Educational and Health Psychology, University College London, London, UK
[3]Children, Young Adults and Families Directorate, Tavistock and Portman NHS Foundation Trust, London, UK
[4]Gender Identity Development Service, Tavistock and Portman NHS Foundation Trust, London, UK
[5]Department of Statistical Science, University College London, London, UK
[6]Research Department of Primary Care and Population Health, University College London, London, UK
[7]Division of Psychiatry, Faculty of Brain Sciences, University College London, London, UK
[8]Department of Paediatric and Adolescent Endocrinology, University College London Hospitals NHS Foundation Trust, London, UK
[9]Autism Research Centre, Department of Psychiatry, University of Cambridge, Cambridge, UK
[10]Institute of Population Health Sciences, University of Liverpool, Liverpool, UK

**Contributors** EK, PC, RO, RH, TW, RS, GB, SB-C, BY and MK contributed to the conception and design of the protocol. EK is the chief investigator of theLongitudinal Outcomes of Gender Identity in Children study. EK, LS, CL, HS and VR drafted the manuscript. RO and VV provided expertise on statistical analysis. RH provided expertise on health economic analysis. All authors drafted or critically revised the protocol and approved the final version of the manuscript.

**Funding** This work was supported by the National Institute for Health Research (NIHR), Health Services and Delivery Research grant number 17/51/19.

**Disclaimer** The views expressed are those of the authors and not necessarily those of the NIHR or the Department of Health and Social Care.

**Competing interests** None declared.

**Patient and public involvement** Patients and/or the public were involved in the design, or conduct, or reporting, or dissemination plans of this research. Refer to the Methods section for further details.

**Patient consent for publication** Not required.

**Provenance and peer review** Not commissioned; externally peer reviewed.

**ORCID iDs**
Eilis Kennedy http://orcid.org/0000-0002-4162-4974
Veronica Ranieri http://orcid.org/0000-0003-0528-8640

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
