## [Reviewer comments · BMJ Open]

ARTICLE DETAILS

TITLE (PROVISIONAL)	Longitudinal Outcomes of Gender Identity in Children (LOGIC): protocol for a prospective longitudinal cohort study of children referred to the UK gender identity development service
AUTHORS	Kennedy, Eilis; Spinner, Lauren; Lane, Chloe; Stynes, Hannah; Ranieri, Veronica; Carmichael, Polly; Omar, Rumana; Vickerstaff, Victoria; Hunter, Rachael; Wright, Talen; Senior, Robert; Butler, Gary; Baron-Cohen, Simon; Young, Bridget; King, Michael

VERSION 1 – REVIEW

REVIEWER	Ashley, Florence University of Toronto
REVIEW RETURNED	23-Oct-2020

GENERAL COMMENTS	Although various points require attention, the protocol is overall clear and the study design appropriate. I recommend acceptance with major revisions as some of the concerns I identify may require further inquiry depending on the authors' responses, notably with regards to consent and ethics. At page 9, lines 17-22, the authors mention that “predictors of individual gender and psychosexual developmental pathways are not clear”. The authors should explain the relevance of prediction if any to establishing best clinical approaches, as various experts and clinicians working in trans health have called that assumption into question in recent years (e.g. 10.1080/15532739.2018.1456390; 10.1177/1359104519836462; 10.1016/j.jadohealth.2015.04.004). This also relates to the study's aim of how to “predict ongoing gender dysphoria and treatment choice”. In the section on recruitment and consent (p. 11), the authors should explain the relationship between the research study and the GIDS clinical care team. Is there any overlap between the teams? How do the researchers ensure that no potentially identifiable information regarding study participants (including who chose to participate or not) is shared between the research and clinical care teams, that participation in the study cannot impact individuals' access to clinical care, and that potential participants understand that refusing/accepting will have no impact on the clinical care they will receive at GIDS? (Concerns of this kind are common in trans health and should be addressed in line with the guidance from Adams et al. at p. 170; 10.1089/trgh.2017.0012). The authors should specify in the section on recruitment and consent whether parents will be allowed participate against the wish of their child, especially where the child is older and/or is
--

	legally competent to consent to research participation and medical care without parental permission or knowledge. At page 12, the authors describe the measures they are seeking. The authors should explain their choice to rely on standardized questionnaires such as the Utrecht Gender Dysphoria Scale, given studies that show that they fail to adequately capture trans experiences amongst adults (10.1007/s10508-019-01556-2) and are likely to mischaracterise the participants' experiences. Table 1 also does not specify how gender identity will be 'measured' and whether this will involve a questionnaire or merely patient/parental reported identity. At pages 12-13, the authors should specify whether the patients' clinicians/endocrinologists will be able to know whether the information was requested for specific patients. If the answer is yes, authors should consider whether it is possible to make arrangements with the clinic to mask that information from clinicians to ensure that participation in the study cannot bias care. At page 13, the authors should mention whether the data will ever be made available to individuals outside of the listed authors, e.g. through a secondary use of data scheme. The section on data storage (p. 16) suggests that this might be the case. If yes, authors should specify whether consent will be obtained in this regard. In the section on sample size, the authors should detail the rationale and empirical basis, if any, for their estimate of 20% attrition rates. Were those based on experiences from previous GIDS studies? Are the estimates conservative and if not, how will the team adapt if they are unable to recruit as many participants as they hope despite recruiting from incoming referrals as well? With regards to powering and regression analysis, the authors should specify whether the study is powered for statistical significance with or without multiple testing correction. If they intend to correct their significance threshold to accommodate for multiple comparisons, they should specify which correction method they will use.
--	---

REVIEWER	Butler, Catherine University of Bath, Psychology
REVIEW RETURNED	02-Dec-2020

GENERAL COMMENTS	This is a much needed study that is long overdue. I was impressed by the wide range of people involved in the study, including an expert in autism. The complexity of this population is captured in the introduction with individual differences such as the presence or absence of autism, gender dysphoria, and social transition, as well the increase of referrals from those assigned female at birth and that a proportion of those referred identify as gender non-binary. However, the focus on individual factors described in the analysis seems to be on autism, early social transition and gender dysphoria and the other factors such as gender assigned at birth do not seem to be considered. It would be important to investigate all aspects of individual differences. When it comes to what is considered an outcome, the introduction describes mental health, self harm / suicidality, alleviating gender dysphoria, wellbeing, psychosexual and gender pathways, access
---

	to medical treatment and the physical and psychological impact of medical treatment. The suggested measures cover most of these outcomes, although it would be important to include psychosexual pathways as it was not clear where this would be covered. Peer and family relationships are described in the aims (but not in the introduction) but there is less emphasis on this in the measures used, the authors might want to consider the SCORE as another measure of this. The sample size required for the study appears to have been calculated based on predicting the number of participants. Was a G-Power calculation also done? This is suggested but the actual calculation is not reported. It is a strength of the study that the authors will also be examining the characteristics of those who drop out of the study.
--	--

REVIEWER	Kuper, Laura University of Texas Southwestern Medical School, Psychiatry
REVIEW RETURNED	21-Dec-2020

GENERAL COMMENTS	The authors are collecting a large, unique, and longitudinally oriented data set that will be important to obtaining information about how to best provide gender affirming care to transgender CYP. Clarification of aims, further description of the GIDS clinic and the larger LOGIC study, refinement of key constructs (gender identity, gender expression, social transition, use of puberty suppression), and greater prioritization of integrating diverse CYP and parent/families into the research process will be needed to strengthen the impact of this important contribution to the literature. Describing limitations of any research findings will be important, particularly within the context of continued evolution in how gender and dysphoria is conceptualized. How does the recent Bell vs Tavistock ruling, which severely limits access to gender affirming medical treatment prior to age 16, impact the current study? Will the authors be able to collect and provide data the speaks to the impact of this decision? Introduction and aims: A number of the physical health measurements are briefly listed but seem outside of the scope of the current protocol. They may be better represented in a separate or sub- protocol (e.g., body composition, bone density). These variables are not well incorporated into the aims and analysis plan. Similarly, no rationale is given for examining costs associated with care. What do the authors seek to learn by collecting this information and how can it be used to improve care? The authors mention higher levels of mental health difficulties and autism among CYP, but it is not clear whether they are referring to samples of younger youth (<14) older youth, or both. Are there any differences by age that would be helpful to mention (e.g., difficulties increase with age)? Given the current protocol references a 2 year follow up period, a number of gaps within the literature that the authors discussed in the introduction cannot be adequately captured. For example, 2 years will only allow for a relatively shorter term snapshot of each
---

	CYP's gender development and possible experiences with puberty suppression. These limitation should be more clearly acknowledged in the introduction. Will any of the CYP be starting estrogen or testosterone therapy during the study period? Better identifying the needs of CYP presenting for care would be a more affirming approach than focus on a model that predicts future medical treatment use. Methods: What is the scope of the larger LOGIC-Q protocol and how can this protocol be accessed? What qualitative data are they collecting? It would be helpful for the authors to provide a brief overview so that the reader has this context without needing to obtain and review other documents. Are you able to collect data on reasons why CYP/families don't attend any appointments? More information on the referral and waitlist process would provide helpful context: How long is the waitlist? How is length of wait and time on treatment factored into the analyses? Will the authors examine the impact of wait time on mental health and gender dysphoria? The authors also mention self-reported tanner staging. Can they compare tanner stage from the first time point to when youth begin receiving treatment? Noting the impact of wait time on mental health, dysphoria, and pubertal development has important implications for how care is provided and is relevant to the ethics of current approaches that require longer waiting times due to lengthier assessment processes that can feel burdensome to CYP/families. The authors mention baseline recruitment of "incoming referrals" and increasing the scope to include those on the waitlist. Do the authors mean that in addition to including families as they are added to the waitlist they will also contact families who were on the waitlist prior to the start of recruitment? The authors description of dividing the waitlist into thirds and how/where this variable is included in the analyses is unclear. How do referrals get made (e.g., can a family reach out directly, do they need a referral from a mental health provider and/or pediatrician?), and what are the criteria for a referral being "accepted"? How do the measures administered in the study compare to those completed as standard of care and when are these standard of care measures administered? The authors discuss extracting physical health data from patient records, could any of the other types of data be extracted as well (gender development, mental health)? The authors discuss gender dysphoria, gender identity, and social transition steps as a key variables, but do not clarify how they are conceptualizing and operationalizing these constructs. I am very concerned that the authors discuss using logistic regression in these analyses, meaning that dysphoria will be treated as a binary
--	---

	variable. It is well established within the field that dysphoria exists on a spectrum and it's not clear that any of the measures listed are meant to be scored in a binary fashion. Any binary cutoff will be arbitrary and not likely to provide meaningful clinical guidance. The authors report that the proportion of youth that they estimate will experience gender dysphoria is 40%. How is this measured, particularly given gender dysphoria exists on a spectrum? Will there be a large enough sample of CYP who progress through puberty without blockers to compare to those with blockers? How will under age 12 and 12 and older measures be compared if they are administered differently, particularly if one version is used at one time point and a different one used at another? Table 1 would benefit from a brief description of what each assessment measure is attempting to capture (e.g., gender identity, expression, dysphoria, mental health and the impact of dysphoria, potential benefits associated with gender affirmation in childhood) and how each measure is scored (scales, subscales, free text qualitative analysis). Many of the gender related measures being collected have been critiqued by transgender researchers and by transgender adults participating in online surveys as poor methods of capturing transgender and non-binary people's experiences of dysphoria. It will be important to reference this literature as well when disseminating findings associated with gender identity and experiences of dysphoria. The mental health and quality of life measures (e.g., YSR, MFQ, mental wellbeing scale) are partially overlapping in content. Which variables from each measure will be potentially included in the analyses? The authors description of statistical tests is fairly vague. My main concern is the high likelihood of false positive results given the potential large number of analyses that could be run with a fairly large sample size that can detect small effects. Can the authors include effect sizes? The authors report 2 follow ups will be conducted but to not explain how each will be included in the analyses. The authors report: "Further subgroup analyses will explore resource use and cost by medical treatment, social transition, co-occurring autistic traits and pre-pubertal children." – why are these important figures to obtain? Is there a "cost" associated with social transition? Could the analyses instead look at per patient/family average costs within the GIDS service and associated cost savings of providing gender affirming care due to reduced mental health burden (e.g., preventing inpatient psychiatric hospitalizations). The authors report: "the research proposal and application for funding which was shared and further reviewed with three parents of young people accessing the service. " --This is a very small number to involve in
--	--

	the research process particularly given the diversity of transgender experiences and the large number of families engaged within the program. Why were CYP left out of the process? Many CYP could be interested in participating. Prioritizing ongoing community involvement and engagement will strengthen the goals, aims, quality of research data, and helpfulness of findings to be used to improve quality of care.
--	---

VERSION 1 – AUTHOR RESPONSE

Response to reviewers: bmjopen-2020-045628

Reviewers' Comments

Reviewer 1:

1. At page 9, lines 17-22, the authors mention that “predictors of individual gender and psychosexual developmental pathways are not clear”. The authors should explain the relevance of prediction if any to establishing best clinical approaches, as various experts and clinicians working in trans health have called that assumption into question in recent years (e.g. 10.1080/15532739.2018.1456390; 10.1177/1359104519836462; 10.1016/j.jadohealth.2015.04.004). This also relates to the study’s aim of how to “predict ongoing gender dysphoria and treatment choice”.

Thank you for highlighting this. The focus of our study is on children and younger adolescents. The statement referred to is from a paper which highlighted a lack of research around pathways and the importance of focusing on experiences from early childhood through to early adulthood in ongoing and future research (Olson-Kennedy et al., 2015). Examining factors associated with gender dysphoria and treatment choice will be helpful in ensuring that children and young people receive appropriate and tailored support. For clarity, the term “predictor” has been replaced with “factors associated with” throughout the paper.

2. In the section on recruitment and consent (p. 11), the authors should explain the relationship between the research study and the GIDS clinical care team. Is there any overlap between the teams? How do the researchers ensure that no potentially identifiable information regarding study participants (including who chose to participate or not) is shared between the research and clinical care teams, that participation in the study cannot impact individuals’ access to clinical care, and that potential participants understand that refusing/accepting will have no impact on the clinical care they will receive at GIDS? (Concerns of this kind are common in trans health and should be addressed in line with the guidance from Adams et al. at p. 170; 10.1089/trgh.2017.0012).

Thank you for this suggestion. We agree with the reviewer that informing participants that their decision to take part in this study will not impact on the clinical care they receive is incredibly important: this is made clear in the participant information sheets. The following information has now been added to the ‘Recruitment and consent’ section of the manuscript:

“Participants will be informed that their decision to participate in the study (or not) will have no impact on their access to care, their position on the referral waiting list, or their future care pathway. Any clinical care team that the participant may go on to have will not know that the individual is taking part in the study unless they choose to share that information with them. Participation in the study will not be routinely documented on the participant’s medical record. The only exception to this is if risk is identified for the child or young person, in which case the research team will follow a risk protocol which may result in a referral to the GIDS enquiries line to initiate support for the family. This information would be documented on the individual’s medical record, but referrals such as this will only be made with the parent/caregiver’s consent. Participants will be informed that they have the

right to withdraw from the study at any time, without providing a reason, and without any impact on their access to care or their care pathway.”

Members of the LOGIC research team who will be leading participant recruitment and data collection, and the study’s CI, are based in the Research & Development Unit at the Tavistock & Portman NHS Foundation Trust, independent of the GIDS. The wider LOGIC team consists of academics based at UCL, the University of Liverpool, the University of Cambridge, and two co-investigators based at the GIDS/UCLH. An administrator in GIDS initially contacts families in line with our ethically-approved recruitment protocol which states that families must first be contacted by a member of staff from the service to which the CYP has been referred. The LOGIC research team is then only permitted to contact those who consent to their contact details being passed on. Therefore, the LOGIC research team does not have access to lists of all those eligible to take part in the study, only to those who have given permission for the team to contact them. The research team does not have access to the medical records system, and information regarding participation (or not) in the study will not be documented on individuals’ medical records. A member of the clinical care team would only be aware that an individual is taking part in the study if the family chose to share that information with them, or if they had been referred to the GIDS enquiries line for support following identified risk for the CYP as part of the study (and even then, only with permission from the parent/caregiver).

Participants will be informed via the information sheets that their decision to participate in the study will not affect their position on the referral waiting list or their future access to care or care pathway.

3. The authors should specify in the section on recruitment and consent whether parents will be allowed participate against the wish of their child, especially where the child is older and/or is legally competent to consent to research participation and medical care without parental permission or knowledge.

The population we are recruiting from is the GIDS waiting list, therefore, it is always the parent/caregiver who is approached about the study by the GIDS administrator or the LOGIC research team as, at baseline, a clinical care team and contact with the CYP has not yet been established. Unfortunately, it is unlikely that we would know of instances where the CYP does not wish for their parent/caregiver to participate as the research team would only have contact with a CYP prior to speaking with the parent/caregiver if they contact the research team directly after receiving a letter about the study (prior to being called by the GIDS administrator). The ethically-approved consent process utilised in this study states that all CYP, regardless of their age, will only be permitted to participate if their parent/caregiver provides informed consent, and they assent. However, parents/caregivers are able to take part in the study alone if their child does not wish to participate. If the research team were made aware that parent/caregiver participation was against a CYP’s wishes, then we would not seek participation from the parent/caregiver.

4. At page 12, the authors describe the measures they are seeking. The authors should explain their choice to rely on standardized questionnaires such as the Utrecht Gender Dysphoria Scale, given studies that show that they fail to adequately capture trans experiences amongst adults (10.1007/s10508-019-01556-2) and are likely to mischaracterise the participants’ experiences. Table 1 also does not specify how gender identity will be ‘measured’ and whether this will involve a questionnaire or merely patient/parental reported identity.

We understand and agree with these concerns regarding the UGDS. However, the UGDS is a validated measure of gender dysphoria which we felt was important to include in the study as it allows us to have some comparability with previous studies (Steensma et al., 2013). A GIDS PPI group of CYP was consulted when the measures were being developed, and whilst some members took issue with some of the statements included in the scale, others were not concerned. Therefore, on balance, we decided to include it in the Young Person questionnaire. We have now included an additional sentence in the instructions for this measure which states: “We are aware that some of these

statements may appear strange and difficult to answer, or you may feel they do not apply to you. Please take all the time you need, ask us questions if you're unsure and try to answer them as best you can."

In order to more fully capture CYP's experiences, CYP will also report on their own gender identity via the child interview (for <12 year olds) or the Young Person questionnaire (for 12+ year olds), and parents/caregivers will also be asked to report their child's gender identity.

5. At pages 12-13, the authors should specify whether the patients' clinicians/endocrinologists will be able to know whether the information was requested for specific patients. If the answer is yes, authors should consider whether it is possible to make arrangements with the clinic to mask that information from clinicians to ensure that participation in the study cannot bias care.

Thank you for highlighting this. The following information has now been added to page 11: "Clinicians and/or Endocrinologists will not know who is taking part in the study or for which participants this information is requested. These data will be collected by a member of the LOGIC research team with assistance from a research nurse at each site."

6. At page 13, the authors should mention whether the data will ever be made available to individuals outside of the listed authors, e.g. through a secondary use of data scheme. The section on data storage (p. 16) suggests that this might be the case. If yes, authors should specify whether consent will be obtained in this regard.

There are no immediate plans to make the data available to individuals outside of the listed authors, but if we were to do so then all relevant regulatory guidelines would be adhered to, including the acquisition of consent from participants, if required.

7. In the section on sample size, the authors should detail the rationale and empirical basis, if any, for their estimate of 20% attrition rates. Were those based on experiences from previous GIDS studies? Are the estimates conservative and if not, how will the team adapt if they are unable to recruit as many participants as they hope despite recruiting from incoming referrals as well?

Unfortunately, as there are so few longitudinal cohort studies in this field, it was not possible to base this estimate on previous similar studies. We think that this estimate is reasonably conservative. All efforts will be made to meet the recruitment target. Participants will be sent birthday cards, newsletters with project updates, and six-monthly check-in emails to maintain engagement with the study, and the study website will be updated as milestones are reached and publications become available. Participant recruitment and retention strategies will be discussed with the PPI group and the Study Steering Committee.

Furthermore, only 200 participants are required to estimate the association between the explanatory variables and the continuous clinical outcomes (the number of participants to estimate 20 regression coefficients with adequate precision is $10 \times 20 = 200$) and we are confident that we will have complete follow-up data on 200 participants. For the binary gender dysphoria outcome, if we are unable to have the required 510 participants with complete follow-up data, (80% of the target 638 participants) we will limit the number of explanatory variables that will be included in the multivariable regression model. This decision will be based on the clinical importance/relevance of the variables and this will be pre-specified in the statistical analysis plan. We will also perform univariable regression analysis. This has now been made clearer in the manuscript.

8. With regards to powering and regression analysis, the authors should specify whether the study is powered for statistical significance with or without multiple testing correction. If they intend to correct their significance threshold to accommodate for multiple comparisons, they should specify which correction method they will use.

The sample size calculation was based on estimating the regression coefficients with adequate precision, rather than power and significance tests, as described in this reference: Peduzzi P, Concato J, Kemper E, Holford TR, Feinstein AR. A simulation study of the number of events per variable in logistic regression analysis. *Journal of Clinical Epidemiology*. 1996 Dec 1;49(12):1373–9. Our primary analyses will use regression models to estimate regression coefficients (measuring the association between the explanatory variables and the outcome) with adequate precision. We will present these estimates with 95% confidence intervals. P-values will not be our focus and therefore, multiple comparisons will not be of concern.

Reviewer 2:

1. This is a much needed study that is long overdue. I was impressed by the wide range of people involved in the study, including an expert in autism. The complexity of this population is captured in the introduction with individual differences such as the presence or absence of autism, gender dysphoria, and social transition, as well the increase of referrals from those assigned female at birth and that a proportion of those referred identify as gender non-binary. However, the focus on individual factors described in the analysis seems to be on autism, early social transition and gender dysphoria and the other factors such as gender assigned at birth do not seem to be considered. It would be important to investigate all aspects of individual differences.

Thank you for these positive comments. Individual differences, including birth-assigned gender, will be examined via exploratory analyses. Table 1 describes all of the variables which we will be collecting data on.

2. When it comes to what is considered an outcome, the introduction describes mental health, self harm / suicidality, alleviating gender dysphoria, wellbeing, psychosexual and gender pathways, access to medical treatment and the physical and psychological impact of medical treatment. The suggested measures cover most of these outcomes, although it would be important to include psychosexual pathways as it was not clear where this would be covered. Peer and family relationships are described in the aims (but not in the introduction) but there is less emphasis on this in the measures used, the authors might want to consider the SCORE as another measure of this.

Additional information has now been added to Table 1 regarding the measurement of psychosexual pathways and peer and family relationships. In the questionnaire for Young People (YP), completed by those aged ≥ 12 years, there are three questions which examine sexuality. The Child version of the questionnaire does not ask about sexuality. Peer and family relationships are examined extensively in all three versions of the questionnaire. The Kidscreen-52 and the Child & Youth Resilience Measures are administered to CYP of all ages. These are standardised measures which have been used in many other child wellbeing studies that will allow us to compare our findings to other studies. Parent/caregivers are also asked about their relationship to the child (e.g. parent from birth, adoptive parent etc.) and how supportive they are of their child's gender identity.

3. The sample size required for the study appears to have been calculated based on predicting the number of participants. Was a G-Power calculation also done? This is suggested but the actual calculation is not reported. It is a strength of the study that the authors will also be examining the characteristics of those who drop out of the study.

The sample size was calculated based on achieving adequate precision for the estimates of regression coefficients and not for predicting the number of participants. It is based on a simple rule of 10 as described below (Peduzzi et al., 1996):

1. For binary outcomes, the number of events (i.e., the number of CYP with gender dysphoria) required in the study is 10 times the number of coefficients to be estimated by the regression model.

2. For continuous outcomes, the number of CYP required is 10 times the number of regression coefficients to be estimated by the regression model.

We have presented our final sample size calculation of 510 (80% of 638, assuming 20% attrition) assuming the proportion of CYP with gender dysphoria to be 40%, which will allow us to estimate 20 regression coefficients with adequate precision using a logistic regression model. However, for the continuous clinical outcomes we only require 200 CYP with complete follow-up. This has now been made clearer in the manuscript.

Reviewer 3:

1. The authors are collecting a large, unique, and longitudinally oriented data set that will be important to obtaining information about how to best provide gender affirming care to transgender CYP.

Clarification of aims, further description of the GIDS clinic and the larger LOGIC study, refinement of key constructs (gender identity, gender expression, social transition, use of puberty suppression), and greater prioritization of integrating diverse CYP and parent/families into the research process will be needed to strengthen the impact of this important contribution to the literature. Describing limitations of any research findings will be important, particularly within the context of continued evolution in how gender and dysphoria is conceptualized.

Thank you for these helpful comments. We agree that these are important aspects to consider and we hope that we have addressed these sufficiently below. We also agree that discussing limitations of the study findings will be important and these will be discussed in future papers, with full consideration of the ever-evolving field and context in which this study is taking place.

2. How does the recent Bell vs Tavistock ruling, which severely limits access to gender affirming medical treatment prior to age 16, impact the current study? Will the authors be able to collect and provide data that speaks to the impact of this decision?

It is likely that the recent ruling will impact on the study. We will collect data as planned and we will endeavour, where possible, to take account of the impact of this decision. Furthermore, the qualitative study will explore the impact of the decision in more detail.

Introduction and aims:

3. A number of the physical health measurements are briefly listed but seem outside of the scope of the current protocol. They may be better represented in a separate or sub-protocol (e.g., body composition, bone density). These variables are not well incorporated into the aims and analysis plan.

Data pertaining to physical health is routinely collected by the endocrine service. The research team will only access this data for those who are a) referred to endocrinology by the end of the study, and b) provide consent for us to do so. It is important to examine physical health outcomes, in particular to identify any untoward adverse physical effects of hormone treatment. These data will be reported descriptively.

4. Similarly, no rationale is given for examining costs associated with care. What do the authors seek to learn by collecting this information and how can it be used to improve care?

We are examining costs associated with care in order to explore how the organisation and resourcing of healthcare might improve outcomes for young people. This research is led by a research team independent of those involved in service delivery and the purpose of our research is to generate knowledge that will be of direct benefit to CYP accessing services. The aim of the analysis is to provide information to policy makers and service providers on the costs and benefits of providing GID services within a finite budget and is based on best practice as set out by national decision making bodies such as The National Institute for Health and Care Excellence (NICE). We have added a paragraph in the introduction (page 6) to clarify this.

5. The authors mention higher levels of mental health difficulties and autism among CYP, but it is not clear whether they are referring to samples of younger youth (<14) older youth, or both. Are there any differences by age that would be helpful to mention (e.g., difficulties increase with age)?

Thank you for noting this. The studies mentioned in the introduction included CYP of a wide age range but we have now mentioned that prevalence of mental health difficulties tends to increase with age. Differences by age will also be explored in our cohort.

6. Given the current protocol references a 2 year follow up period, a number of gaps within the literature that the authors discussed in the introduction cannot be adequately captured. For example, 2 years will only allow for a relatively shorter term snapshot of each CYP's gender development and possible experiences with puberty suppression. These limitation should be more clearly acknowledged in the introduction. Will any of the CYP be starting estrogen or testosterone therapy during the study period?

We agree that two years is a relatively short follow-up period, especially considering the length of time CYP must wait before their first appointment with the service. The study has been funded for a 2-year follow-up in the first instance, but we will seek further funding to extend this so that we can better capture gender development, possible experiences with puberty suppression, and treatment pathways and wellbeing over time. Limitations regarding follow-up will be discussed fully in subsequent papers. We anticipate that some CYP will start cross-sex hormone treatment during the study period.

7. Better identifying the needs of CYP presenting for care would be a more affirming approach than focus on a model that predicts future medical treatment use.

We completely agree that identifying the needs of CYP presenting for care is of utmost importance. We hope that the broad range of measures included in the study, as well as the qualitative arm of the study, will allow for an in depth exploration and understanding of this.

Methods:

8. What is the scope of the larger LOGIC-Q protocol and how can this protocol be accessed? What qualitative data are they collecting? It would be helpful for the authors to provide a brief overview so that the reader has this context without needing to obtain and review other documents.

A separate protocol for the LOGIC-Q study has been submitted to BMJ Open and is currently under review. A reference to this has now been included in the protocol. We originally submitted all aspects of the LOGIC programme of research within one protocol but were asked by the journal to resubmit these as separate protocols. All protocols will be available on the LOGIC study website once published. The qualitative study involves semi-structured interviews with a purposively sampled subset of up to 40 families who have completed the quantitative assessments. Additional information about the qualitative study has now been included in the protocol.

9. Are you able to collect data on reasons why CYP/families don't attend any appointments?

The study hopes to capture the experiences of families who decide to leave the waitlist or the service, so information will be collected at each time point to determine whether families are still on the waitlist. If they are no longer on the waitlist, families will be asked to indicate whether this is because their child is now undergoing assessment with the service or whether the family have decided to leave the waitlist, prior to having an initial appointment with the service. In this instance, families will be asked to indicate why they chose to leave the waitlist. Data relating to appointments will not be

routinely collected as part of this study. However, this will be explored within LOGIC-Q, where appropriate.

10. More information on the referral and waitlist process would provide helpful context. How long is the waitlist? How is length of wait and time on treatment factored into the analyses? Will the authors examine the impact of wait time on mental health and gender dysphoria? The authors also mention self-reported tanner staging. Can they compare tanner stage from the first time point to when youth begin receiving treatment? Noting the impact of wait time on mental health, dysphoria, and pubertal development has important implications for how care is provided and is relevant to the ethics of current approaches that require longer waiting times due to lengthier assessment processes that can feel burdensome to CYP/families.

Further information regarding the referral and waitlist process has now been included in the manuscript. Based on information from the GIDS website, the service do not provide an exact figure for the length of the waitlist as there are a number of factors which influence this (e.g. the increase in referrals in recent years). However, the website states that the service are currently seeing CYP who were referred in 2018. Data pertaining to length of wait and time on treatment will be included in the regression models. Length of wait and time on treatment will be compared descriptively between those who are lost to follow-up and the completers in subsequent papers. Tanner stage data will be collected at each time point so it will be possible to compare tanner stage from the first time point to when youth begin receiving treatment.

11. The authors mention baseline recruitment of “incoming referrals” and increasing the scope to include those on the waitlist. Do the authors mean that in addition to including families as they are added to the waitlist they will also contact families who were on the waitlist prior to the start of recruitment? The authors description of dividing the waitlist into thirds and how/where this variable is included in the analyses is unclear.

Yes, the study is recruiting both families who have just been referred to the service (incoming referrals) as well as those currently on the waitlist. In order to be eligible to participate in the study, families must still be on the waitlist when they complete the baseline measures. Thank you for highlighting that our description is unclear, this has now been amended.

12. How do referrals get made (e.g., can a family reach out directly, do they need a referral from a mental health provider and/or pediatrician?), and what are the criteria for a referral being “accepted”?

Referral to the service can be made by Child and Adolescent Mental Health Services (CAMHS), or from other health, social care or educational professionals, including GPs. The service has a referral form which the referrer completes. The service does not accept self-referrals from CYP or their families. Referrals are screened by an intake administrator and then reviewed and discussed by clinicians in the intake team. The referral must indicate that the CYP is experiencing concerns or distress around their gender identity in order to be accepted. Further information may be requested from the referrer if this information is not clear from the referral form.

13. How do the measures administered in the study compare to those completed as standard of care and when are these standard of care measures administered? The authors discuss extracting physical health data from patient records, could any of the other types of data be extracted as well (gender development, mental health)?

There is some overlap between the measures administered in the study and those completed as standard of care. The only parent measure that overlaps is the CBCL. For CYP aged 12 years and older, the measures that overlap are the YSR, UGDS and GDQ. These measures are all completed

as standard of care at baseline (within the first three appointments at the service), 6-8 months after baseline and then twice yearly. For CYP who opt for the early intervention pathway, the parent and CYP also complete the Kidscreen-52. All of the measures are completed after one year on GnRHa and after one year on CSH. Apart from these measures, all of the measures administered in the study are specific to the study and not completed as standard of care. Although it would be possible to extract data from patient records, we have opted to collect all data directly from the CYP and their family as this is a more reliable approach than extracting information from clinical records. We can then ensure that all measures are completed and that these are completed at the correct time point.

14. The authors discuss gender dysphoria, gender identity, and social transition steps as a key variables, but do not clarify how they are conceptualizing and operationalizing these constructs. I am very concerned that the authors discuss using logistic regression in these analyses, meaning that dysphoria will be treated as a binary variable. It is well established within the field that dysphoria exists on a spectrum and it's not clear that any of the measures listed are meant to be scored in a binary fashion. Any binary cutoff will be arbitrary and not likely to provide meaningful clinical guidance.

Gender dysphoria is assessed using DSM5 criteria and participants indicate the extent to which they agree or disagree with each of the statements (e.g. "I want to be treated as another gender to the one I was given at birth"), on a 5 point Likert scale. It is therefore possible to treat gender dysphoria as either a binary or a continuous variable. We have calculated our sample size conservatively treating gender dysphoria as a binary variable. However, it will also be possible for us to investigate associations between the pre-specified explanatory variables and severity of gender dysphoria (treated as a continuous variable). We have now provided further information regarding the conceptualisation and operationalisation of our variables in Table 1.

15. The authors report that the proportion of youth that they estimate will experience gender dysphoria is 40%. How is this measured, particularly given gender dysphoria exists on a spectrum?

40% experiencing gender dysphoria is an estimate based on clinician experience and clinical data. As noted above, gender dysphoria is assessed using DSM5 criteria and participants indicate the extent to which they agree or disagree with each of the statements, on a 5 point Likert scale. It will therefore be possible for us to investigate the proportion of CYP with gender dysphoria, as well as severity of gender dysphoria.

16. Will there be a large enough sample of CYP who progress through puberty without blockers to compare to those with blockers?

We anticipate that we will have a large enough sample to be able to compare CYP who progress through puberty without blockers to CYP who have blockers. It is difficult for us to predict exactly what the sample sizes of these groups will be. Furthermore, external factors (e.g. the recent Bell vs. Tavistock judicial review) will likely have an impact. We will certainly be mindful of these factors when interpreting the study findings.

17. How will under age 12 and 12 and older measures be compared if they are administered differently, particularly if one version is used at one time point and a different one used at another?

Parental measures will be completed at all time points and the majority of the measures are the same for all age groups so it will be possible to compare these data over time for those who are younger than 12 years at baseline but older than 12 years at follow-up. For CYP, some of the measures are the same across age groups so it will be possible to compare these. We will take into account differences in measures for CYP who complete different measures at different time points.

18. Table 1 would benefit from a brief description of what each assessment measure is attempting to capture (e.g., gender identity, expression, dysphoria, mental health and the impact of dysphoria, potential benefits associated with gender affirmation in childhood) and how each measure is scored (scales, subscales, free text qualitative analysis).

Thank you for highlighting this. We have now included additional information in Table 1.

19. Many of the gender related measures being collected have been critiqued by transgender researchers and by transgender adults participating in online surveys as poor methods of capturing transgender and non-binary people's experiences of dysphoria. It will be important to reference this literature as well when disseminating findings associated with gender identity and experiences of dysphoria.

Thank you for noting this, we completely agree and will include discussion of the limitations of these measures when presenting our findings. We have included a range of both standardised measures and bespoke measures in order to address these limitations, whilst ensuring comparability between studies. This is certainly something that will be discussed further in our subsequent papers.

20. The mental health and quality of life measures (e.g., YSR, MFQ, mental wellbeing scale) are partially overlapping in content. Which variables from each measure will be potentially included in the analyses?

Thank you for highlighting this important point. The variables that will be included in our analyses will be described in our detailed statistical analysis plan, prior to analysing the data.

21. The authors description of statistical tests is fairly vague. My main concern is the high likelihood of false positive results given the potential large number of analyses that could be run with a fairly large sample size that can detect small effects. Can the authors include effect sizes?

Our primary analyses will use descriptive summary measures and regression models. The estimates of the regression coefficients measuring the association between the explanatory variables and the outcomes will be reported with their corresponding 95% confidence intervals. Our conclusions will not focus on P-values so we will therefore not be at risk of false positives.

22. The authors report 2 follow ups will be conducted but to not explain how each will be included in the analyses.

Models that account for repeated measurement data will be used as appropriate, including time period as an additional covariate. This is now mentioned in the manuscript. A more detailed statistical analysis plan will be produced before analysis and database lock.

23. The authors report: "Further subgroup analyses will explore resource use and cost by medical treatment, social transition, co-occurring autistic traits and pre-pubertal children." – why are these important figures to obtain? Is there a "cost" associated with social transition? Could the analyses instead look at per patient/family average costs within the GIDS service and associated cost savings of providing gender affirming care due to reduced mental health burden (e.g., preventing inpatient psychiatric hospitalizations).

The sub-groups chosen are pre-specified analyses for groups where there is a potential difference in costs and benefits, as is recommended by national decision making bodies such as the National Institute for Health and Care Excellence (NICE) in England. Children with co-occurring autistic traits, for example, may have a higher level of need, and hence a higher cost of care. This is important

information for policy makers and service providers when addressing the needs of these complex groups. For children that socially transition there is the potential that there is no difference in costs, but this in itself would be important information to know. Children that socially transition may also have a different benefit from contacts with GIDS compared to those who do not.

Our current analyses will report the average cost per child of GIDS as well as the average cost per child of mental health care contacts. On page 14 we have included the text “numbers, percentages, means and standard deviations per CYP” to make this clearer. Children with contact with GIDS will be compared with those with no GIDS contacts as is specified on page 14. It is important to note that GIDS in itself may increase the cost of mental health care by facilitating contacts with services, for example through referrals to other services following discharge from care. As a result we are unable to specifically look at “cost savings”, particularly given that this an observational cohort. What we will examine instead is if increased costs also result in increased quality of life, which would suggest that children access services that appropriately address their needs.

24. The authors report: “the research proposal and application for funding which was shared and further reviewed with three parents of young people accessing the service. “ --This is a very small number to involve in the research process particularly given the diversity of transgender experiences and the large number of families engaged within the program. Why were CYP left out of the process? Many CYP could be interested in participating. Prioritizing ongoing community involvement and engagement will strengthen the goals, aims, quality of research data, and helpfulness of findings to be used to improve quality of care.

When drafting the research proposal, research priorities were discussed with both CYP and their families at service user days for GIDS service users. We therefore incorporated the views of both CYP and their families in our research design. We then shared the full research proposal and application with three parents of young people accessing the service in order to obtain detailed feedback on the proposal. Throughout the study period, we will meet regularly with our study patient and public involvement group to seek ongoing feedback and advice. This group is comprised of both CYP and their families who are participating in the study, so we will ensure that all views are represented. The co-investigator team also includes an academic who has experience of service use as a young person.

VERSION 2 – REVIEW

REVIEWER	Ashley, Florence University of Toronto
REVIEW RETURNED	18-Feb-2021

GENERAL COMMENTS	The authors have adequately responded to my concerns. I recommend acceptance. I only offer a small recommendation for their consideration, which they are free to take or leave: given authors' preference not to include parents whose CYP do not consent to their participants (addressed in query #3), they may wish to systematically ask parents whether their children are aware of and/or opposed to their participation. This could highlight opposition that would not otherwise come to light and otherwise provide useful information when comparing the CYP+parent and parent-only subgroups.
---

REVIEWER	Butler, Catherine University of Bath, Psychology
-----------------	---

REVIEW RETURNED	09-Mar-2021
GENERAL COMMENTS	This is a much needed and timely study. The authors have addressed the revisions requested and it should be accepted as it stands.